# MANICOG: TRAINING-FREE IMPROVEMENT FOR GUI GROUNDING VIA MANIPULATION CHAINS

## ABSTRACT

GUI grounding is a critical capability for enabling GUI agents to execute tasks such as clicking and dragging. However, in complex scenarios like the ScreenSpot-Pro benchmark, existing models often suffer from suboptimal performance. Utilizing the proposed **Masked Prediction Distribution (MPD)** attribution method, we identify that the primary sources of errors are twofold: high image resolution (leading to precision bias) and intricate interface elements (resulting in ambiguity bias). To address these challenges, we introduce the **Manipulation-based Chain of GUI Grounding (ManiCoG)**, which incorporates two key manipulations, coarse-to-fine focus and candidate selection, to effectively mitigate these biases. Our extensive experimental results demonstrate that ManiCoG significantly enhances the accuracy of various GUI grounding models in a training-free setting. For instance, applying our method to the TianXi-Action-7B model boosts its accuracy on the ScreenSpot-Pro benchmark from 51.9% to 57.8%. Furthermore, ablation studies confirm the robustness of the ManiCoG approach across diverse parameter configurations, highlighting its stability and effectiveness.

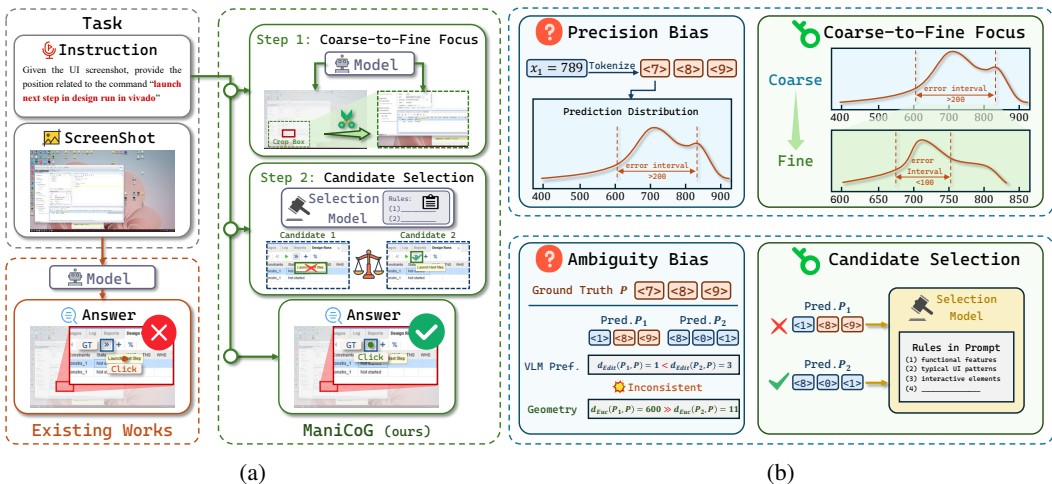

Figure 1: **Overview.** (a) Compared with conventional grounding models, **ManiCoG** achieves accurate localization without additional training via a manipulation chain. (b) To address accuracy bias and ambiguity bias, ManiCoG introduces two manipulations: coarse-to-fine focus and candidate selection.

## 1 INTRODUCTION

The advent of multimodal large language models (MLLMs) (Hurst et al., 2024; Bai et al., 2025) has made it increasingly feasible for GUI agents to automate tasks across desktop and mobile platforms. At the core of these agents lies *GUI Grounding*: given a pair of *natural language instructions* and a *screenshot*, the task is to accurately localize the coordinates of the target element within a high-resolution graphical interface, thereby enabling subsequent atomic actions such as clicking, typing, or dragging. Early approaches often relied on structured interface representations, such as XML or DOM trees (Deng et al., 2023; Gur et al., 2024). However, these structures are

frequently unavailable or inconsistent with the visual rendering in real-world scenarios. Consequently, research has shifted toward the visual paradigm of *instruction + screenshot*, where MLLMs directly output coordinates (Wu et al., 2024; Xu et al., 2024; Lu et al., 2024; Gou et al., 2025; Qin et al., 2025), providing a more robust perceptual foundation for agents. In comparison to general natural image tasks, GUI scenarios present unique challenges due to their **high resolution** and **dense elements**, where semantics are determined by a combination of icons, text, and contextual cues. These characteristics make accurate localization significantly more challenging. For instance, in ScreenSpot-Pro (Li et al., 2025), a benchmark dataset covering professional software across multiple domains, the localization accuracy of most models remains below 50%.

The performance of multimodal grounding models remains underutilized. Notably, improvements in performance can be achieved **without additional training** by optimizing inference methods. From an error-driven perspective, we categorize grounding failures into two primary types: **(1) Knowledge deficiency**: The model fails to recognize the target due to a lack of relevant knowledge. **(2) Inductive bias**: The model has the necessary knowledge but makes errors due to its inherent selection bias, which manifests in two typical forms, namely *precision bias* and *ambiguity bias*. To diagnose these causes of failure, we introduce a **Masked Prediction Distribution (MPD)** method. This approach randomly occludes parts of the screenshot, makes repeated predictions, and aggregates the frequency of hotspots or candidate points across the image. This aggregation reveals how the model distributes its focus across the image. Statistical analysis of 50 error samples shows that approximately 14% of failures stem from knowledge deficiency, while 74% are attributed to inductive bias.

In this paper, we propose the **Manipulation-based Chain of GUI Grounding (ManiCoG)**. The key idea is to transform the one-step localization task into a recursive, multi-step chain of reasoning through predefined manipulations (Fig. 1). To mitigate precision bias, we decompose localization into hierarchical **coarse-to-fine focus**, where each step refines the candidate region identified in the previous round. This progressive refinement reduces the search space and improves the resolution of the predicted coordinates. To address ambiguity bias, we incorporate an external **Candidate Selection**. By defining selection rules specific to the localization task and injecting these rules into the model as prompts, we correct the model's erroneous selection preferences. Importantly, our method does not require any additional model training and can be directly applied to a variety of existing

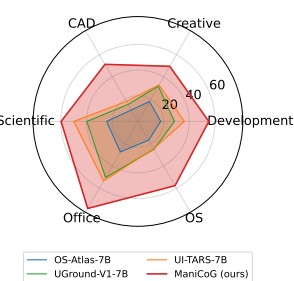

Figure 2: Accuracy comparison on ScreenSpot-Pro.

open-source backbones. We evaluate ManiCoG on multiple open-source backbones (e.g., OS-Atlas-7B (Wu et al., 2024), UI-TARS-7B (Qin et al., 2025), and TianXi-Action-7B (Tang et al., 2025b)) and several datasets (e.g., ScreenSpot-Pro (Li et al., 2025), ScreenSpot-V2 (Wu et al., 2024)). ManiCoG consistently improves accuracy on complex samples (Fig. 2). Ablation studies further confirm the effects of **coarse-to-fine focus** and **candidate selection**. Our results demonstrate that extending and structuring the reasoning path during inference provides a cost-effective means of unlocking the full grounding potential of existing models. The main contributions of this work are as follows:

- **Diagnosis of Grounding Failures**: We introduce the MPD method to diagnose common grounding failures, such as knowledge deficiency and inductive bias.
- **Precision Bias Mitigation**: We transform single-step localization into a multi-step progressive search through hierarchical cropping, which effectively reduces precision bias in high-resolution and small-object scenarios.
- **Ambiguity Bias Correction**: To address discrepancies between MLLM's edit distance and spatial coordinate distance, we introduce an external selection and correct the MLLM's selection bias using predefined rules injected as prompts.
- **Training-free Improvements**: We validate ManiCoG across various backbones and benchmarks, demonstrating consistent improvements and emphasizing the general value of test-time reasoning design in GUI Grounding.

## 2 RELATED WORK

Training on pre-trained MLLMs (Bai et al., 2025) has been demonstrated to significantly enhance GUI grounding capabilities. Early approaches predominantly relied on conventional instruction fine-tuning. With the introduction of DeepSeek-R1 (Guo et al., 2025), reinforcement learning fine-tuning has

attracted growing attention. Meanwhile, several studies have found that specially designed inference methods help tap into the potential of MLLMs in terms of localization capabilities.

## 2.1 INSTRUCTION FINE-TUNING

The simplest approach is to fine-tune pre-trained MLLMs (e.g., Qwen2.5-VL (Bai et al., 2025)) on task-specific GUI instruction datasets. Early work such as AGUVIS (Xu et al., 2024) introduced vision-based models for GUI grounding. To address high-resolution GUI screenshots, CogAgent (Hong et al., 2024) introduced a cross-resolution efficient attention mechanism. ShowUI (Lin et al., 2025) applied token pruning based on GUI interface structure, improving both efficiency and performance. OmniParser (Lu et al., 2024) converted GUI pixels into structured tokens that could be parsed by LLMs. In terms of dataset construction, SeeClick (Cheng et al., 2024) proposed an automated pipeline for managing GUI data. UGround (Gou et al., 2025) built a large-scale dataset with 10M elements, improving generalization. With the advent of larger-scale datasets and more powerful models, new large-scale systems such as UI-TARS (Qin et al., 2025) and Phi-Ground (Zhang et al., 2025b) have pushed the state-of-the-art performance across various benchmarks.

## 2.2 REINFORCEMENT LEARNING

Given the fine-grained nature of GUI localization, instruction fine-tuning alone is often insufficient for achieving high precision. DeepSeek-R1 (Guo et al., 2025) introduced the GRPO method, demonstrating the potential of reinforcement learning in enhancing spatial reasoning for GUI grounding tasks. Following this, UI-R1 (Lu et al., 2025) and GUI-R1 (Luo et al., 2025) were among the first to apply GRPO in GUI tasks. InfiGUI-R1 (Liu et al., 2025) focused on reward function design, emphasizing IoU-based metrics to improve localization accuracy. GUI-G1 (Zhou et al., 2025) introduced box-attribute constraints to regulate bounding-box geometry, while GUI-G2 (Tang et al., 2025a) modeled spatial distributions using Gaussian functions. TianXi-Action (Tang et al., 2025b) focused on generating high-quality reinforcement learning data. Collectively, these studies affirm the efficacy of reinforcement learning in enhancing spatial reasoning and fine-grained prediction in GUI tasks.

## 2.3 INFERENCE ENHANCEMENT

Significant attention has been given to optimizing inference strategies to fully exploit the capabilities of MLLMs. One line of work extends reasoning chains in the language space; however, experiments (Zhang et al., 2025a) have found this direction suboptimal for GUI scenarios, sometimes even hindering performance. Alternatively, several works have targeted inference enhancement in the image space. ScreenSeekeR (Li et al., 2025) and R-VLM (Park et al., 2025) introduced multi-stage pipelines, first performing region-level localization followed by refinement within local regions, thus improving accuracy. DiMo-GUI (Wu et al., 2025) proposed a divide-and-conquer strategy, separating reasoning over icons and text to reduce cross-modal interference. GUI-RC (Du et al., 2025) employed intersection operations to aggregate multiple predictions, improving robustness. While conventional MLLMs have demonstrated the effectiveness of inference enhancement techniques for general tasks (Liu et al., 2024), their direct application to GUI tasks is often limited by inductive biases specific to spatial reasoning. This paper identifies two critical inductive biases —**precision bias and ambiguity bias**— that remain prominent in GUI grounding. We propose the ManiCoG framework to address these issues through a manipulation-chain design.

## 3 PILOT STUDY

On ScreenSpot-Pro (Li et al., 2025), a challenging GUI grounding benchmarks, the accuracy of state-of-the-art grounding models on these benchmarks has significantly decreased, falling below 50%. To gain deeper insights into the underlying performance bottlenecks, we conducted a systematic pilot study addressing two primary questions: *(1) What are the root causes of errors made by GUI grounding models? (2) How can these errors be mitigated from a model mechanism perspective without the need for retraining?*

### 3.1 ERROR ATTRIBUTION: MASKED PREDICTION DISTRIBUTION

This section uses the ScreenSpot-Pro dataset (Li et al., 2025) as a benchmark to analyze potential error patterns in GUI grounding models and explore corresponding mitigation strategies.

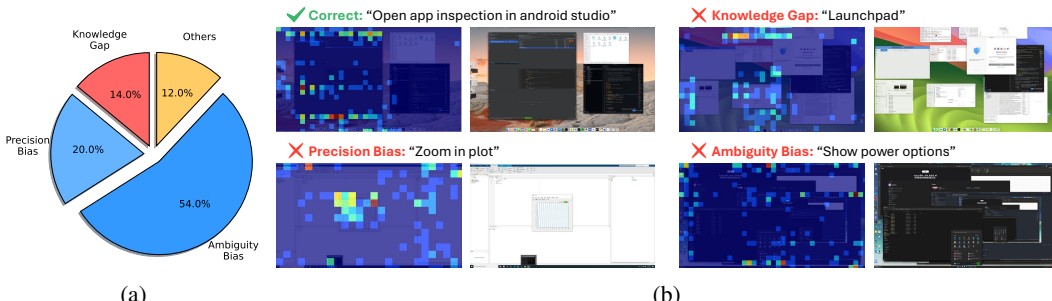

Figure 3: Error Attribution Analysis. (a) Proportions of attribution types. (b) Attribution analysis of model predictions. The deep red regions in the heatmap indicate potential prediction locations, demonstrating how the MPD method can clearly identify the sources of model errors.

**Problem Formulation**   For a GUI grounding model $f$, given a query $q$ and a GUI screenshot $I \in \mathbb{R}^{H \times W \times 3}$, the model generates a text sequence $t$ containing the target bounding box in the standard format: `<|box_start|>`$(x_1, y_1, x_2, y_2)$`<|box_end|>`. The coordinates $(x_1, y_1, x_2, y_2) = r(t)$ are extracted using a regular expression parser $r$, where $(x_1, y_1)$ and $(x_2, y_2)$ represent the top-left and bottom-right coordinates of the bounding box, respectively. The center coordinates of the bounding box are computed as: $(x_c, y_c) = \left( \frac{x_1 + x_2}{2}, \frac{y_1 + y_2}{2} \right)$. A prediction is considered correct if the center coordinate $(x_c, y_c)$ lies within the ground-truth bounding box; otherwise, it is deemed an error.

**Attribution Method**   Traditional gradient-based attribution methods (e.g., GradCAM (Selvaraju et al., 2017), Integrated Gradients (Sundararajan et al., 2017)) are not well-suited for the discrete text-to-coordinate conversion process. As an alternative, we initially considered using Shapley values (Shapley et al., 1953; Lundberg & Lee, 2017) for attribution analysis. For an $n$-dimensional input feature, the Shapley value for the $i$-th feature is defined as: $\phi_i = \sum_{S \subseteq \{1,2,...,n\} \setminus \{i\}} \frac{|S|!(n-|S|-1)!}{n!} [f(S \cup \{i\}) - f(S)]$, where $S$ denotes a subset of features. However, due to the high resolution of GUI screenshots, estimating the Shapley values (Ancona et al., 2019) for a single sample takes approximately 10 hours on a single RTX 4090 GPU, which is computationally impractical. To address this, we propose the **Masked Prediction Distribution (MPD)** method, which efficiently observes the spatial distribution patterns of model predictions under random perturbations (see the detailed MPD procedure in appendix Algorithm 2). Regions with densely distributed predicted points indicate high model confidence in those areas. We set the number of perturbations to 300 per sample and can obtain the MPD heatmap within 20 minutes per sample.

**Error Pattern Analysis**   Based on the experimental results of TianXi-Action-7B (Tang et al., 2025b) on ScreenSpot-Pro, we conducted an attribution analysis on 50 error samples, with the findings summarized in Table 1. Notably, both precision bias and ambiguity bias are categorized as inductive bias issues, collectively accounting for 74% of the error samples. This indicates that if we can effectively mitigate inductive bias, the model's performance will be significantly improved.

### 3.2 MITIGATION STRATEGY: INDUCTIVE BIAS CORRECTION

Based on the error pattern analysis, we explored potential mitigation methods for different error types. Knowledge gap errors reflect limitations in the model's training data or architecture, which are difficult to address with inference-time techniques. In contrast, inductive bias errors (precision bias and ambiguity bias) can potentially be mitigated through optimization of the inference mechanism.

**Limitations of Language-Space Enhancement**   Inspired by reasoning techniques in large language models (e.g., Chain-of-Thought (Wei et al., 2022)), we first attempted to enhance GUI grounding performance by augmenting linguistic information. **(1) Query Expansion Strategy:** For queries with insufficient or ambiguous descriptions, we used a language model to expand and refine the original query, generating more precise instruction information. **(2) Context Expansion Strategy:** We utilized a multimodal large language model (e.g., Qwen2.5-VL (Bai et al., 2025)) to generate a structured description of the GUI, including the geometric location, text content, and other information

Table 1: The proportions and detailed analysis of different error types.

| Error Type | Description and Analysis |
|---|---|
| Knowledge Gap (14%) | Attributions indicate that the model fails to recognize information related to the ground-truth bounding box, with predicted point distribution showing no correlation with the target area. This category includes 7 error samples, stemming from the model's insufficient ability to recognize specific UI elements or interaction patterns. |
| Precision Bias (20%) | The model correctly identifies the target region but exhibits systematic offset between predicted and ground-truth boxes. Attribution results show predicted points densely distributed near the ground-truth box with shifted center positions. 10 samples belong to this error type. |
| Ambiguity Bias (54%) | While successfully identifying information from the ground-truth box, the model is simultaneously distracted by other similar regions, leading to predictions oriented toward incorrect areas. Attribution reveals multiple clusters of predicted points. 27 erroneous samples fall into this category, making it the most prevalent error type. |
| Others (12%) | The remaining 6 error samples belong to unclassified error patterns. |

of UI elements, and concatenated this with the original query as model input. However, experimental results indicated that merely extending the language sequence did not significantly improve model accuracy, and even introduced additional errors in some cases. This phenomenon aligns with recent findings (Zhang et al., 2025a) that traditional linguistic reasoning models are difficult to directly transfer to precise grounding tasks.

**Root Causes of Precision Bias** An in-depth analysis of precision bias revealed that multimodal models typically adopt discretized coordinate representations for images with resolution $H \times W$. For instance, in Qwen series models, a coordinate value of $x_1 = 789$ is split into independent digit characters (<7>, <8>, <9>) and further converted into their corresponding token IDs. This discretization inherently limits the model's maximum precision to the unit digit level.

**Root Causes of Ambiguity Bias** The cross-entropy training objective for multimodal models optimizes the **edit distance** of token sequences rather than the **Euclidean distance**. Let the ground-truth coordinate be $x_{\text{GT}} = 789$, and consider two predicted candidates: $x' = 189$ and $x'' = 801$. A direct comparison of the two metrics yields:

- Edit distance: $d_{\text{edit}}(x_{\text{GT}}, x') = 1 < d_{\text{edit}}(x_{\text{GT}}, x'') = 3$
- Euclidean distance: $d_{\text{euc}}(x_{\text{GT}}, x') = 600 > d_{\text{euc}}(x_{\text{GT}}, x'') = 12$

This inconsistency in metrics causes a fundamental conflict between the model's optimization objective in token space and the need for accuracy in real-world spatial localization. Therefore, external correction mechanisms combining token sequence optimization with geometric constraints are necessary to address this systematic bias.

## 4 METHOD

Based on the experimental results from the pilot study, we design the **ManiCoG** method in this section. The method targets both accuracy bias and ambiguity bias, and proposes different manipulations to improve the accuracy and robustness of GUI grounding.

### 4.1 ACCURACY BIAS ELIMINATION: COARSE-TO-FINE FOCUS

The root cause of accuracy bias lies in the discretization process of multimodal large models during coordinate localization. Since the prediction accuracy of the model is typically limited to the pixel level, and its output is difficult to be perfectly accurate, prediction errors may sometimes reach tens or even hundreds of pixels. Therefore, to effectively eliminate accuracy bias, inspired by human observation strategies, we propose a **coarse-to-fine focus** manipulation. Specifically, we first use the grounding model to predict a coarse localization coordinate $(x^t, y^t)$. Then, based on this coarse coordinate, we crop the original image to a scale of $\lambda < 1$, and input the cropped image back into the grounding model for fine localization, obtaining a more precise coordinate $(x^{t+1}, y^{t+1})$. Although this process can be iterated multiple times, we find that there is a trade-off in the hyperparameters. **(1) Iteration count:** After a certain number of iterations, the performance improvement of the model

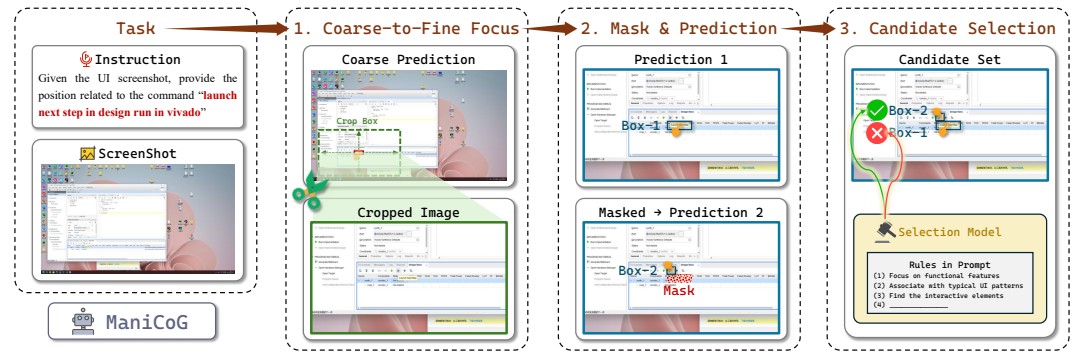

Figure 4: Illustration of ManiCoG. **Step 1:** Based on the initial prediction results of the grounding model, ManiCoG performs cropping around these initial predictions at a predefined ratio. **Step 2:** The model conducts multiple predictions on the cropped images; after each prediction, the pixels within the predicted bounding box are randomly masked to ensure the diversity of multiple prediction results. **Step 3:** Using predefined rules and an external knowledge model, the model ranks multiple candidate coordinates and selects the final coordinates for output.

---

**Algorithm 1** ManiCoG (with $N$ crop iterations and $M$ candidates per iteration)

---

**Require:** Query $q$, screenshot $I$, correction model $m$, and grounding model $f$
**Ensure:** Grounding point $(x, y)$
  1: Initialize the input image as $I^1 = I$
  2: **for all** $t \in \{1, 2, \cdots, N\}$ **do**
  3:     Initialize the candidate box set $\Phi^t = \emptyset$
  4:     **for all** $i \in \{1, 2, \cdots, M\}$ **do**
  5:         Masking all pixels in the candidate set to get input image $I_i^t = \text{MASK}(I^{t-1}, \Phi^t)$
  6:         Predict the candidate box $b_i^t = f(q, I_i^t)$ and update $\Phi^t \leftarrow \Phi^t \cup \{b_i^t\}$
  7:     **end for**
  8:     Select the preferred box $\tilde{b}^t = m(q, I^t, \Phi^t)$
  9:     Crop the input image $I^{t+1} = \text{CROP}(I^t, \tilde{b}^t)$
 10: **end for**
 11: Compute the center point of $\tilde{b}^N$ as $(x, y)$

---

tends to plateau; **(2) Crop ratio:** A large cropping ratio may lead to the loss of crucial information, while a small cropping ratio may prevent the model from accurately localizing the target.

### 4.2 AMBIGUITY BIAS ELIMINATION: CANDIDATE CORRECTION

Multimodal large models (MLLMs) represent coordinates as text sequences for autoregressive generation. While this design simplifies the training process, it introduces a discrepancy between the training and inference phases. For example, the coordinate "789" is encoded into the text sequence <7><8><9>, and the model minimizes the edit distance of this text sequence using cross-entropy loss. In practice, however, the impact of digit position errors is asymmetric: an error in the hundreds place is two orders of magnitude more significant than that in the ones place. This results in a substantial mismatch between edit distance and Euclidean distance, and no straightforward mapping exists to convert the former to the latter. To eliminate ambiguity bias, we first generate multiple mutually exclusive candidate bounding boxes through multi-round masked prediction operations. Subsequently, we utilize an external correction model (e.g., GPT-5) to re-select from these candidate boxes. Notably, the key to this operation lies in prompt design; as shown in the results of Table 5, naive prompt designs fail to leverage the correction model effectively. To enable the correction model to rectify the erroneous ordering tendency of the grounding model, we incorporate **key principles** consistent with GUI priors into the prompt. Examples of these principles are provided below, and detailed prompt design is available in Section C.1.

```
────────────────────────── Prompt ──────────────────────────
1    - (Functional Preference) Focus on the functional purpose of the highlighted elements
2    - (Memory Comparison) Consider standard patterns (e.g., buttons for actions)
3    - (Interactive Components) Prioritize interactive elements over static text/labels
```

### 4.3 ManiCoG: Manipulation-based Chain of GUI Grounding

By integrating the two manipulations outlined above, we propose **ManiCoG**, as illustrated in Fig. 4. To enhance the diversity of candidate boxes, we mask the pixels within already predicted candidate boxes prior to each new prediction step, thereby ensuring the mutual exclusivity between newly generated candidate boxes and existing results. To mitigate precision bias, ManiCoG adopts a coarse-to-fine focus strategy in its outer loop, enabling gradual refinement of focus toward more accurate coordinate positions step by step. Simultaneously, to address ambiguity bias, ManiCoG employs a candidate selection strategy in each iteration, selecting the most suitable box from multiple candidates as the final output. The algorithmic implementation of ManiCoG is detailed in Algorithm 1.

## 5 Experiment

### 5.1 Experimental Setup

**Models**   The proposed ManiCoG method aims to enhance the accuracy of Grounding models without retraining. We tested this method on several state-of-the-art grounding models, including OS-Atlas-7B (Wu et al., 2024), UI-TARS-1.5-7B (Qin et al., 2025), and TianXi-Action-7B (Tang et al., 2025b). All models were implemented using the Transformers framework (Wolf et al., 2019) for inference. The input to the models consists of both the query and the screenshot. OS-Atlas and TianXi-Action output bounding box coordinates, while UI-TARS outputs click coordinates.

**Data**   We evaluate ManiCoG on ScreenSpot-V2 (Wu et al., 2024), and ScreenSpot-Pro (Li et al., 2025). ScreenSpot-V2 are mainly used to assess grounding accuracy in simple scenarios, covering mobile, web, and desktop. ScreenSpot-Pro focuses on complex scenarios, consisting of high-resolution screenshots of professional software, where each sample contains multiple software elements, and the targets are typically small, making it a particularly challenging task.

**Hyperparameters**   To balance efficiency and accuracy, two iterations were adopted for the coarse-to-fine focusing process. For high-resolution screenshots, the crop ratio $\lambda$ was set to the range $[0.5, 0.7]$. To eliminate ambiguity bias, a masking mechanism was employed, which generates $2 \sim 3$ candidate results per iteration; subsequently, an external API (e.g., GPT-5) was used to select the result most relevant to the query. All experiments were conducted on a single RTX 4090 GPU.

Table 2: Comparison with various models on ScreenSpot-Pro.

| Grounding Model | Development | | Creative | | CAD | | Scientific | | Office | | OS | | Avg. |
|---|---|---|---|---|---|---|---|---|---|---|---|---|---|
| | Text | Icon | Text | Icon | Text | Icon | Text | Icon | Text | Icon | Text | Icon | |
| *Proprietary Models* | | | | | | | | | | | | | |
| GPT-4o (Hurst et al., 2024) | 2.0 | 0.0 | 1.3 | 0.0 | 1.0 | 0.0 | 2.1 | 0.0 | 1.1 | 0.0 | 0.0 | 0.0 | 0.8 |
| Claude Computer Use (Hu et al., 2024) | 14.5 | 3.7 | 22.0 | 3.9 | 25.9 | 3.4 | 33.9 | 15.8 | 30.1 | 16.3 | 11.0 | 4.5 | 17.1 |
| *General Open-source Models* | | | | | | | | | | | | | |
| Qwen2.5-VL-3B (Bai et al., 2025) | 9.1 | 7.3 | 22.1 | 1.4 | 26.8 | 2.1 | 38.2 | 7.3 | 33.9 | 15.1 | 10.3 | 1.1 | 16.1 |
| Qwen2.5-VL-7B (Bai et al., 2025) | 16.8 | 1.6 | 46.8 | 4.1 | 35.9 | 7.7 | 49.3 | 7.3 | 52.5 | 20.8 | 37.4 | 6.7 | 26.8 |
| *GUI-specific Models (SFT)* | | | | | | | | | | | | | |
| SeeClick-9.6B (Cheng et al., 2024) | 2.5 | 0.0 | 0.6 | 0.0 | 1.0 | 0.0 | 3.5 | 0.0 | 1.1 | 0.0 | 2.8 | 0.0 | 1.1 |
| CogAgent-18B (Hong et al., 2024) | 7.1 | 3.1 | 14.9 | 0.7 | 9.6 | 0.0 | 22.2 | 1.8 | 13.0 | 0.0 | 5.6 | 0.0 | 7.7 |
| OS-Atlas-7B (Wu et al., 2024) | 12.2 | 4.7 | 33.1 | 1.4 | 28.8 | 2.8 | 37.5 | 7.3 | 33.9 | 5.7 | 27.1 | 4.5 | 18.9 |
| ShowUI-2B (Lin et al., 2025) | 2.5 | 0.0 | 16.9 | 1.4 | 9.1 | 0.0 | 13.2 | 7.3 | 15.3 | 7.5 | 10.3 | 2.2 | 7.7 |
| UGround-7B (Gou et al., 2025) | 14.2 | 1.6 | 26.6 | 2.1 | 27.3 | 2.8 | 31.9 | 2.7 | 31.6 | 11.3 | 17.8 | 0.0 | 16.5 |
| UGround-V1-7B (Gou et al., 2025) | 15.8 | 1.2 | 51.9 | 2.8 | 47.5 | 9.7 | 57.6 | 14.5 | 60.5 | 13.2 | 38.3 | 7.9 | 31.1 |
| UI-TARS-7B (Qin et al., 2025) | 20.8 | 9.4 | 58.4 | 12.4 | 50.0 | 9.1 | 63.9 | 31.8 | 63.3 | 20.8 | 30.8 | 16.9 | 35.7 |
| TianXi-Action-7B (Tang et al., 2025b) | **76.0** | **21.4** | **61.6** | **19.6** | **45.2** | **18.8** | **80.6** | **31.8** | **84.2** | **54.7** | **57.9** | **33.7** | **51.9** |
| *GUI-specific Models (RL)* | | | | | | | | | | | | | |
| UI-R1-3B (Lu et al., 2025) | 11.2 | 6.3 | 22.7 | 4.1 | 27.3 | 3.5 | 42.4 | 11.8 | 32.2 | 11.3 | 13.1 | 4.5 | 17.8 |
| UI-R1-E-3B (Lu et al., 2025) | 37.1 | 12.5 | 46.1 | 6.9 | 41.9 | 4.2 | 56.9 | 21.8 | 65.0 | 26.4 | 32.7 | 10.1 | 33.5 |
| GUI-R1-7B (Luo et al., 2025) | 23.9 | 6.3 | 49.4 | 4.8 | 38.9 | 8.4 | 55.6 | 11.8 | 58.7 | 26.4 | 42.1 | 16.9 | - |
| InfiGUI-R1-3B (Liu et al., 2025) | 33.0 | 14.1 | 51.3 | 12.4 | 44.9 | 7.0 | 58.3 | 20.0 | 65.5 | 28.3 | 43.9 | 12.4 | 35.7 |
| GUI-G1-3B (Zhou et al., 2025) | 39.6 | 9.4 | 50.7 | 10.3 | 36.6 | 11.9 | 61.8 | 30.0 | 67.2 | 32.1 | 23.5 | 10.6 | 37.1 |
| SE-GUI-7B (Yuan et al., 2025) | 51.3 | 42.2 | 68.2 | 19.3 | 57.6 | 9.1 | 75.0 | 28.2 | 78.5 | 43.4 | 49.5 | 25.8 | 47.3 |
| GUI-G2-7B (Tang et al., 2025a) | **55.8** | **12.5** | **68.8** | **17.2** | **57.1** | **15.4** | **77.1** | **24.5** | **74.0** | **32.7** | **57.9** | **21.3** | **47.5** |
| *Test-Time Methods* | | | | | | | | | | | | | |
| GUI-RC (Du et al., 2025) | - | - | - | - | - | - | - | - | - | - | - | - | 41.2 |
| DiMo-GUI-7B (Wu et al., 2025) | 66.9 | 21.4 | 60.6 | 21.7 | 50.3 | 14.1 | 68.1 | 21.8 | 80.8 | 52.8 | 69.2 | 28.1 | 49.7 |
| ManiCoG-7B | **81.8** | **26.9** | **68.2** | **23.8** | **58.4** | **29.7** | **77.8** | **36.4** | **83.6** | **60.4** | **72.9** | **33.3** | **57.8** |

Table 3: Comparison with different baseline models on ScreenSpot-Pro.

| Grounding Model | Development | | Creative | | CAD | | Scientific | | Office | | OS | | Avg. |
|---|---|---|---|---|---|---|---|---|---|---|---|---|---|
| | Text | Icon | Text | Icon | Text | Icon | Text | Icon | Text | Icon | Text | Icon | |
| UGround-7B (Gou et al., 2025) | 14.2 | 1.6 | 26.6 | 2.1 | 27.3 | 2.8 | 31.9 | 2.7 | 31.6 | 11.3 | 17.8 | 0.0 | 16.5 |
| + ManiCoG | **48.7** | **5.5** | **46.5** | **7.7** | **18.3** | **4.7** | **54.9** | **14.6** | **52.5** | **18.9** | **42.8** | **9.4** | **30.0** |
| OS-Atlas-7B (Wu et al., 2024) | 12.2 | 4.7 | 33.1 | 1.4 | 28.8 | 2.8 | 37.5 | 7.3 | 33.9 | 5.7 | 27.1 | 4.5 | 18.9 |
| + ManiCoG | **66.2** | **16.6** | **58.6** | **16.1** | **36.0** | **10.9** | **55.6** | **17.3** | **68.4** | **22.6** | **56.8** | **17.1** | **41.6** |
| UI-TARS-1.5-7B (Qin et al., 2025) | 50.0 | 14.5 | 56.6 | 13.3 | 37.6 | 12.5 | 66.0 | 22.7 | 76.3 | 34.0 | 55.6 | 16.9 | 40.8 |
| + ManiCoG | **71.4** | **22.1** | **68.2** | **21.7** | **49.8** | **14.1** | **77.8** | **23.6** | **82.5** | **41.5** | **69.1** | **24.2** | **51.9** |
| TianXi-Action-7B (Tang et al., 2025b) | 76.0 | 21.4 | 61.6 | 19.6 | 45.2 | 18.8 | 80.6 | 31.8 | 84.2 | 54.7 | 57.9 | 33.7 | 51.9 |
| + ManiCoG | **81.8** | **26.9** | **68.2** | **23.8** | **58.4** | **29.7** | **77.8** | **36.4** | **83.6** | **60.4** | **72.9** | **33.3** | **57.8** |

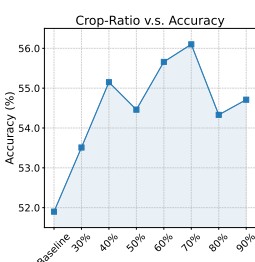 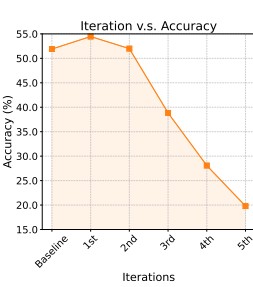 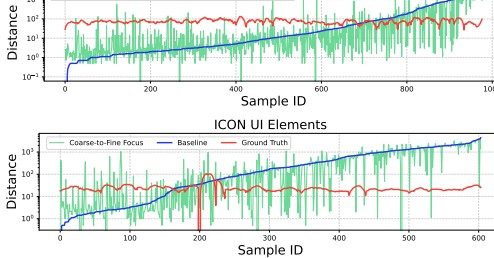

(a) Ablations on crop ratio and iteration number.  (b) Impact of different target types.

## 5.2 COMPARISON WITH SOTA

We first evaluated state-of-the-art grounding methods on the complex ScreenSpot-Pro dataset, with results summarized in Table 2. All models were categorized into three groups based on their training or deployment paradigms: supervised fine-tuning (SFT) training, reinforcement learning (RL) training, and test-time inference. Among 7B-scale models, our ManiCoG method achieved the best performance on the ScreenSpot-Pro dataset, attaining an accuracy of 57.8%. Built upon TianXi-Action-7B (Tang et al., 2025b), our model delivers a 5.9% accuracy improvement. We present the consistent improvements of ManiCoG across different base models in Table 3. Additionally, we conducted experiments on the ScreenSpot-V2 dataset, and detailed results are provided in appendix Table 7, which demonstrates that our method outperforms all baseline models to varying extents. Furthermore, Table 4 dissects two key manipulations of ManiCoG, where "PB Eli." and "AB Eli." denote precision bias elimination and ambiguity bias elimination, respectively. It can be observed that both manipulations independently yield significant improvements in model accuracy.

## 5.3 ABLATION STUDIES

This section presents a series of ablation experiments to validate the effectiveness of the accuracy bias and ambiguity bias elimination manipulations in ManiCoG and explores the impact of different parameter settings on model performance.

### 5.3.1 ACCURACY BIAS ELIMINATION

**Impact of Crop Ratio and Iteration Number**   We first investigate the effects of the number of crops and the crop ratio $\lambda$ on the elimination of accuracy bias, as illustrated in Fig. 5a. For the crop ratio, the number of iterations is fixed at 2. When the crop ratio exceeds 40%, the elimination effect on precision bias becomes relatively significant. If the crop ratio is set too aggressively (i.e., less than 40%), it may lead to the cropping of crucial contextual information, thereby compromising model performance. For the number of iterations, the crop ratio is fixed at 50%. It can be observed that 2 iterations are sufficient to eliminate precision bias. Excessive iterations may result in an overly large overall cropping ratio, which conversely degrades the prediction performance.

**Impact of Target Type**   We also aim to investigate whether ManiCoG exhibits selectivity in its ability to eliminate accuracy bias across different targets, as illustrated in Fig. 5b. We calculated the Euclidean distance between the model's predicted points and the ground truth, where the blue line

represents the baseline and the green line represents ManiCoG. The red line denotes the distance between the corner points and the center point of the ground truth bounding box. If a prediction line lies below the red line, the model's prediction is highly likely to be correct. We sorted the baseline results in ascending order for ease of observation. It can be seen that for both text and icon types, the bias distance of ManiCoG's predictions is mostly smaller than that of the baseline method. This indicates that ManiCoG has no selective preference for predicted targets and possesses universality.

### 5.3.2 AMBIGUITY BIAS ELIMINATION

Table 4: Ablation on proposed manipulations.

| Setting | PB Eli. | AB Eli. | Accuracy |
|---|---|---|---|
| Baseline | | | 51.9 |
| + Coarse-to-Fine Focus | ✓ | | 55.2 |
| + Candidate Selection | | ✓ | 54.3 |
| **+ ManiCoG** | ✓ | ✓ | **57.8** |

Table 5: Ablation on prompt design.

| Setting | CoT | KP | Accuracy |
|---|---|---|---|
| Baseline | | | 51.9 |
| + Vanilla Prompt | | | 55.7 |
| + Prompt w/ CoT | ✓ | | 57.0 |
| **+ Prompt w/ CoT & KP** | ✓ | ✓ | **57.8** |

Table 6: Impact of different correction models.

| Correction Model | Baseline | Doubao-Seed-1.6-Flash | GLM-4.5V | Qwen-VL-Max | Gemini-2.5-Pro | **GPT-5** |
|---|---|---|---|---|---|---|
| **Accuracy** | 51.9 | 55.3 | 55.9 | 56.4 | 57.2 | **57.8** |

**Impact of Prompt Design**  A key reason for ambiguity bias lies in the fact that MLLMs prioritize candidate outcomes based on edit distance. Therefore, it is crucial to inject priority priors in the Euclidean space through prompt design. We conducted ablation experiments on two important prompt structures in ManiCoG, as shown in Table 5. "CoT" denotes the chain-of-thought-style prompt, which aims to enable the correction model to make selections in a more granular manner. "KP" stands for key principle, a critical component for injecting coordinate space priority priors into the selection process. Experimental results demonstrate that injecting Euclidean space priors into the correction model significantly enhances the accuracy of ManiCoG (see Section C.1 for detailed prompts).

**Impact of Correction Model Selection**  We investigated the impact of correction model selection, as presented in Table 6. GPT-5 and Gemini-2.5-Pro achieved the best performance, enabling an overall accuracy of over 57%. All other models also contributed to performance improvement. These results indicate that our ManiCoG method is not sensitive to the selection of correction models.

## 6 DISCUSSION

This paper focuses on investigating how to enhance the GUI grounding task in a training-free manner. First, we propose the **MPD** method to analyze the underlying causes of incorrect predictions in existing GUI grounding models. Based on this analysis, we identify that the majority of incorrect predictions are primarily attributed to two types of biases in the models: **accuracy bias and ambiguity bias**. Through a detailed examination of the causes of these two biases, we design the **ManiCoG** method. This method extends the model's reasoning process by introducing two critical manipulations (i.e., coarse-to-fine focus and candidate selection), which significantly alleviates the aforementioned biases. On ScreenSpot-Pro, a currently challenging benchmark, our method achieves an accuracy of 57.8%, representing a 5.9% improvement over the baseline method TianXi-Action-7B. However, we also identify several limitations of ManiCoG. Due to the adoption of the training-free paradigm, we have to integrate a correction model via an external API, which is unfavorable for users with higher privacy requirements. In future work, we will consider training a lightweight local correction model to replace the reliance on external APIs. Additionally, we find that the most fundamental approach to eliminating the model's inductive bias lies in fully accounting for these two biases during the model training process. We will also explore the theoretical differences between the model's inductive preferences and real-world scenarios to develop more generalizable solutions.

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

## TABLE OF CONTENT FOR APPENDIX

## A   USAGE OF LARGE MODELS IN PAPER WRITING

During the conduct of this research, we utilized the GPT-5 for auxiliary support, primarily encompassing the following two aspects:

- **Manuscript Polishing:** Leveraging the text generation capability of GPT-5, we polished the draft of this manuscript, focusing on correcting grammatical errors, addressing expression inconsistencies, and other related issues. It should be emphasized that all content of the manuscript was still manually composed; the LLM was not involved in formulating the research logic of the paper. Additionally, all text generated by the LLM underwent manual review and revision to ensure its quality and accuracy.

- **Literature Survey:** We employed the knowledge retrieval capability (Retrieval-Augmented Generation, RAG) of GPT-5 to search for relevant literature. To guarantee retrieval accuracy, all retrieved literature was subject to manual review and verification. Subsequently, we screened out literature relevant to the research topic, followed by thorough reading and systematic organization of the selected materials.

## B   DETAILS OF THE PROPOSED METHODS

### B.1   DETAILED ALGORITHM OF MPD ATTRIBUTION

To investigate the root causes of errors in grounding models, we propose a method for rapidly computing the decision attribution of models, namely **Masked Prediction Distribution (MPD) Attribution**. The detailed steps of this algorithm are presented as follows:

---

**Algorithm 2** Masked Prediction Distribution (MPD) Attribution Algorithm

---

**Require:** GUI image $I$, query $q$, grid size $(M, N)$, number of samples $K$
**Ensure:** Set of predicted points $\mathcal{P} = \{(x_c^{(k)}, y_c^{(k)})\}_{k=1}^{K}$
  1: Partition the image $I$ into $M \times N$ grid blocks $\{B_{i,j}\}_{i=1,j=1}^{M,N}$
  2: **for** $k = 1$ to $K$ **do**
  3:   Randomly select a masking ratio $\alpha$ and sample $\lfloor \alpha \cdot M \cdot N \rfloor$ grid blocks to mask
  4:   Generate the masked image $I^{(k)}$, where masked regions are filled with zero vectors
  5:   Compute the model prediction: $t^{(k)} = f(q, I^{(k)})$
  6:   Extract the center coordinates: $(x_c^{(k)}, y_c^{(k)})$
  7: **end for**
  8: Visualize all predicted points $\{(x_c^{(k)}, y_c^{(k)})\}_{k=1}^{K}$ as a scatter plot

---

## C  EXPERIMENTAL DETAILS

### C.1  PROMPT DESIGN

The design of prompts is crucial for injecting prior information of coordinate space into the candidate box selection process. In the experiments presented in Table 5, we compare prompts with different content. Among these, the vanilla prompt is as follows:

```
                                   ── Prompt ──
1  You are comparing two images to determine which one better fulfills the user's intent.
2
3  User Command: "{user_query}"
4
5  Image 1: Shows a GUI element marked with a green box labeled "1"
6  Image 2: Shows a GUI element marked with a red box labeled "2"
7
8  Your task: Determine which image shows the element that will best fulfill the user's
   command.
9
10 **OUTPUT FORMAT**:
11 <answer>1 or 2</answer>"""
```

This simplistic prompt design fails to rectify the model's ambiguity bias. Therefore, in our ManiCoG method, we incorporate two critical structures—chain of thought and key principle—to enhance the model's understanding of prior information regarding the coordinate space. The final prompt we employed is presented as follows:

```
                                   ── Prompt ──
1  You are comparing two images to determine which one better fulfills the user's intent.
2
3  User Command: "{user_query}"
4
5  Image 1: Shows a GUI element marked with a green box labeled "1"
6  Image 2: Shows a GUI element marked with a red box labeled "2"
7
8  Your task: Determine which image shows the element that will best fulfill the user's
   command.
9
10 ANALYSIS APPROACH:
11 1. Examine what GUI element is highlighted in each image
12 2. Consider which element better matches the user's intent
13 3. Think about standard GUI patterns and user expectations
14 4. Choose the image that shows the more appropriate interaction target
15
16 KEY PRINCIPLES:
17 - Focus on the functional purpose of the highlighted elements
18 - Consider standard UI patterns (buttons for actions, text fields for input, etc.)
19 - Choose interactive elements over static text/labels
20 - If one shows a selected state and the other shows normal state, prefer the normal state
21 - ELEMENT QUALITY HIERARCHY (best to worst):
22     - Icon + Text together (most informative and complete)
23     - Complete icon alone (clear visual indicator)
24     - Complete text alone (readable label)
25     - Multiple elements in one box OR incomplete elements (ambiguous target)
26
```

```
27    COMMON PITFALLS TO AVOID:
28        - Don't choose based on keyword matching alone
29        - Don't overlook the user's actual goal in favor of literal interpretation
30
31    Remember: Provide SPECIFIC analysis based on what you actually observe, not generic
      descriptions.
32
33    **OUTPUT FORMAT**:
34    <analysis>
35    Image 1: [Describe what element is highlighted and its purpose]
36    Image 2: [Describe what element is highlighted and its purpose]
37    Comparison: [Explain which better serves the user's intent and why]
38    </analysis>
39
40    <answer>1 or 2</answer>
41    <reason>Brief explanation of why this image shows the better choice</reason>
```

## C.2 MODEL INFERENCE DETAILS

The models employed in this study can be broadly categorized into two types:

- **Bounding box-output models**: Such as OS-Atlas-7B (Wu et al., 2024) and TianXi-Action-7B (Tang et al., 2025b)

- **Click point-output models**: Such as UGround (Gou et al., 2025) and UI-TARS-1.5-7B (Qin et al., 2025)

For **bounding box-output models**, the implementation of masked prediction is straightforward—only the pixels within the output bounding boxes need to be masked. In contrast, for **click point-output models**, we first expand the region around each click point by a fixed number of pixels (e.g., 25 pixels) in the up, down, left, and right directions, and then mask the expanded region.

## D MORE EXPERIMENTS

### D.1 COMPARISON ON SCREENSPOT-V2

In addition to validating the ManiCoG method on the ScreenSpot-Pro (Li et al., 2025) dataset, we also conducted validation on the simpler ScreenSpot-V2 (Wu et al., 2024) dataset. Unlike ScreenSpot-Pro, most grounding models already achieve satisfactory accuracy on ScreenSpot-V2; this is attributed to the lower resolution of samples and the simpler elements contained in individual samples within the latter dataset. When we applied the ManiCoG method to the OS-Atlas-7B and UI-TARS-1.5-7B models, further performance improvements were observed. However, the magnitude of these improvements is smaller than that achieved on the ScreenSpot-Pro dataset.

### D.2 WHY MASKING IS ADOPTED INSTEAD OF RANDOM SAMPLING?

In conventional approaches for generating candidate detection boxes, random sampling is typically employed. Specifically, when predicting the next token, instead of using the `torch.argmax` function to greedily select the token corresponding to the highest score, top-k/top-p sampling methods are utilized to obtain candidate tokens. However, our experiments reveal that in GUI grounding models during candidate box generation, the score difference between the top-1 token and top-2 token is substantial. This directly leads to a significant issue: candidate boxes generated via random sampling tend to cluster in a single region. As illustrated in Fig. 6a, the red boxes represent candidate boxes obtained through random sampling. It is evident that these boxes exhibit almost complete overlap and lack diversity, which renders the subsequent selection process largely meaningless.

To address this limitation, we propose a masking strategy: pixels within the already predicted candidate boxes are masked first. This ensures that subsequently predicted candidate boxes are mutually exclusive with the already predicted ones. As shown in Fig. 6b, the green boxes are candidate boxes generated using the masked prediction method. These boxes demonstrate significantly greater distribution diversity, thereby enhancing the upper performance limit of selection manipulation.

Table 7: Comparison with various models on ScreenSpot-V2.

| Grounding Model | Mobile | | Desktop | | Web | | Avg. |
|---|---|---|---|---|---|---|---|
| | Text | Icon | Text | Icon | Text | Icon | |
| InternVL-2-4B (Chen et al., 2024) | 9.2 | 4.8 | 4.6 | 4.3 | 0.9 | 0.1 | 4.3 |
| Qwen2-VL-7B (Wang et al., 2024) | 61.3 | 39.3 | 52.0 | 45.0 | 33.0 | 21.8 | 42.9 |
| CogAgent (Hong et al., 2024) | 67.0 | 24.0 | 74.2 | 20.0 | 70.4 | 28.6 | 47.4 |
| SeeClick (Cheng et al., 2024) | 78.0 | 52.0 | 72.2 | 30.0 | 55.7 | 32.5 | 53.4 |
| OS-Atlas-4B (Wu et al., 2024) | 85.7 | 58.5 | 72.2 | 45.7 | 82.6 | 63.1 | 70.1 |
| UGround-7B (Gou et al., 2025) | 82.8 | 82.8 | 82.8 | 63.6 | 80.4 | 70.4 | 73.3 |
| OS-Atlas 7B (Wu et al., 2024) | 92.1 | 68.7 | 88.7 | 60.7 | 89.7 | 75.9 | 81.2 |
| **+ ManiCoG** | **92.4** | **67.3** | **88.7** | **66.4** | **89.3** | **79.8** | **82.2** |
| UI-TARS-1.5-7B (Qin et al., 2025) | 94.1 | 80.6 | 88.7 | 76.4 | 88 | 84.2 | 86.4 |
| **+ ManiCoG** | **94.1** | **80.6** | **88.7** | **76.4** | **88** | **84.7** | **86.5** |

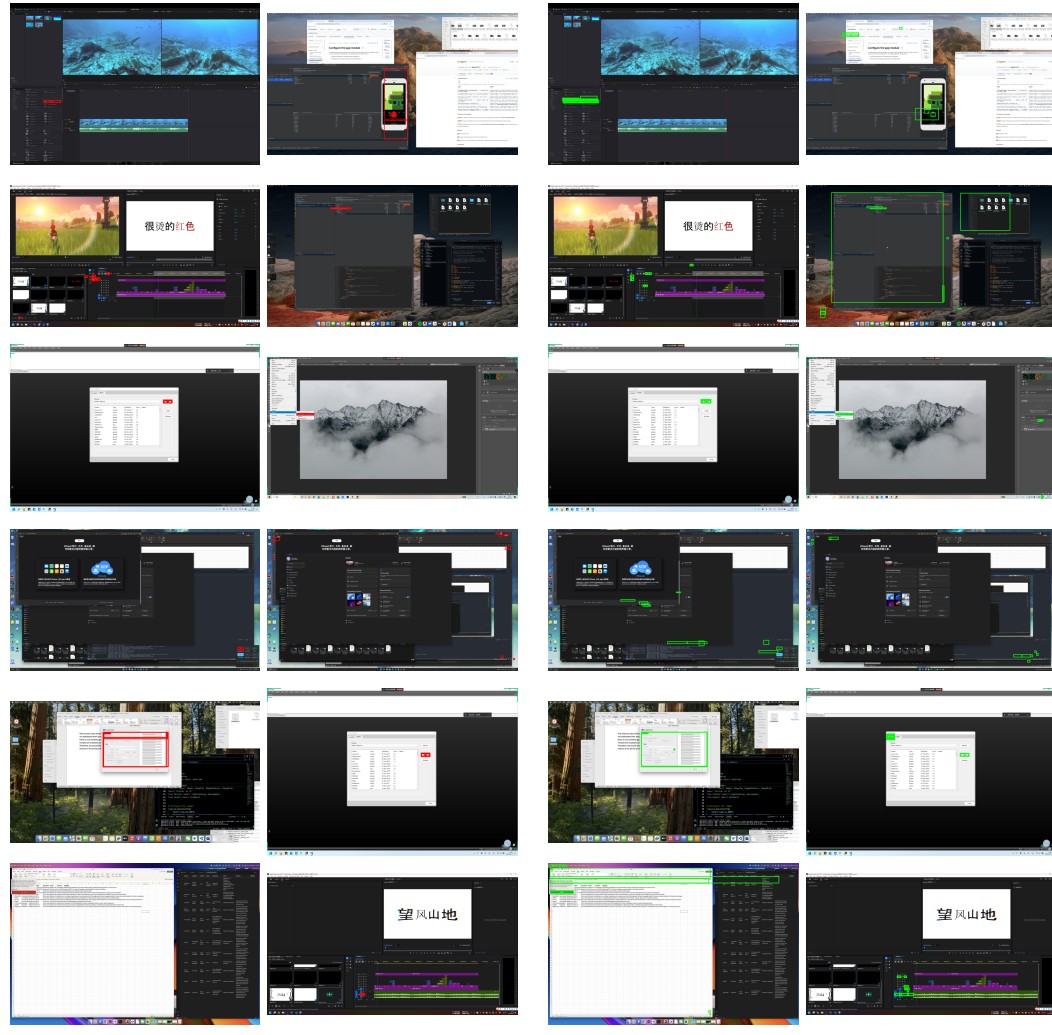

(a) Candidate boxes with random sampling.      (b) Candidate boxes with masked prediction.

### D.3 MORE VISUALIZATIONS OF MANICOG

To better demonstrate the process by which ManiCoG corrects the baseline model, 8 samples were randomly selected from cases where the baseline model made incorrect predictions but ManiCoG achieved accurate corrections, as shown in Fig. 7. In the figure, green boxes represent ground truth, red boxes denote the baseline model's prediction results (incorrect), and blue boxes indicate the corrected results by ManiCoG (correct). Specifically, ManiCoG utilized 2 candidate boxes in each prediction round of this experiment, with the correction model employing GPT-5. In these samples, it can be observed that accurately predicting bounding boxes in accordance with user instructions is considerably challenging, as the figures contain substantial interfering information. By alleviating precision bias and ambiguity bias, ManiCoG successfully achieves correct predictions in these samples.

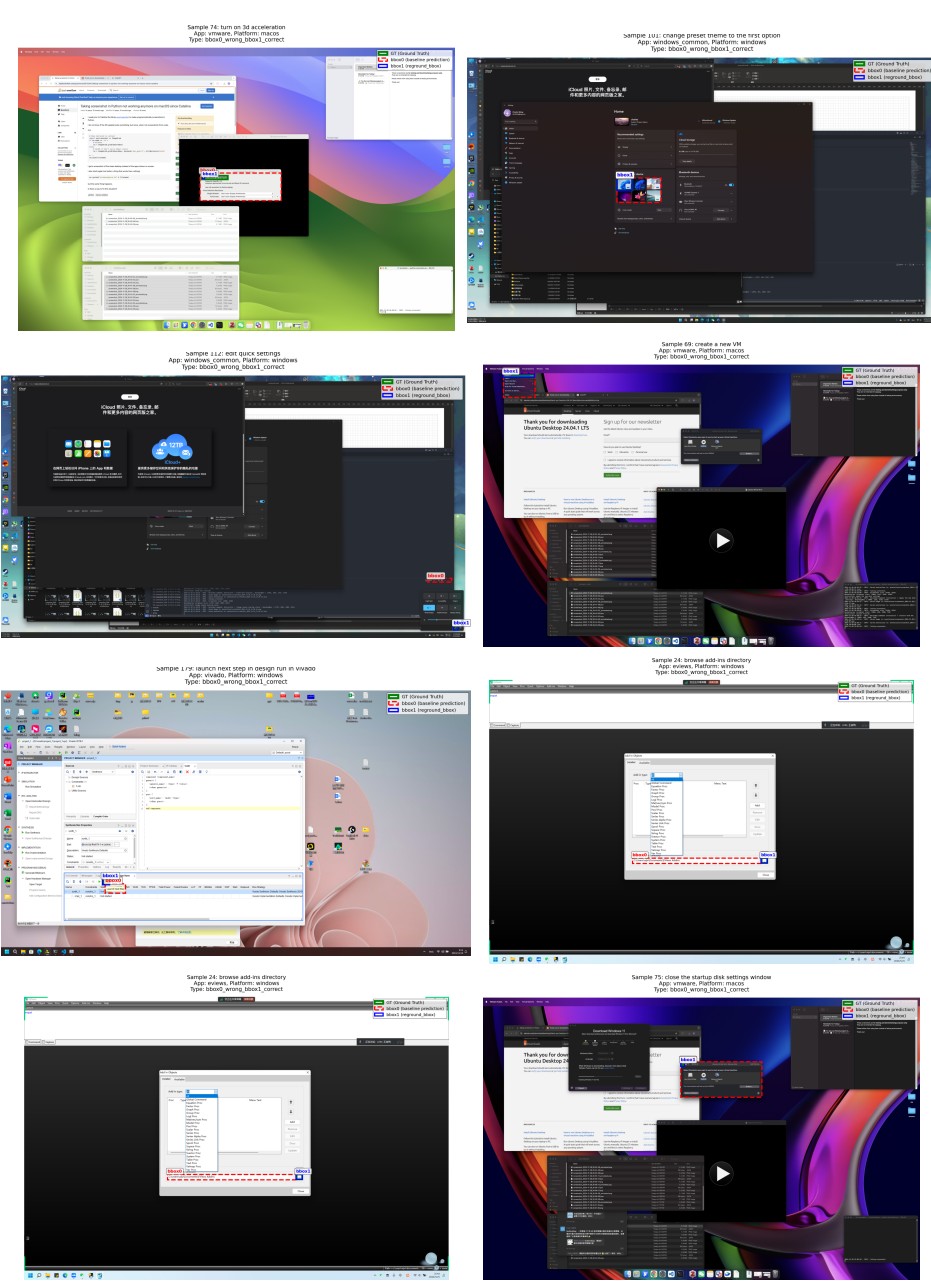

Figure 7: Visualizations of ManiCoG.

### D.4 MORE ATTRIBUTION RESULTS

We present additional attribution results herein to comprehensively demonstrate the attribution capability of the Masked Prediction Distribution (MPD) method. Specifically, we randomly selected samples from four categories (`Correct / Knowledge Gap / Precision Bias / Ambiguity Bias`) as illustrated in Fig. 8.

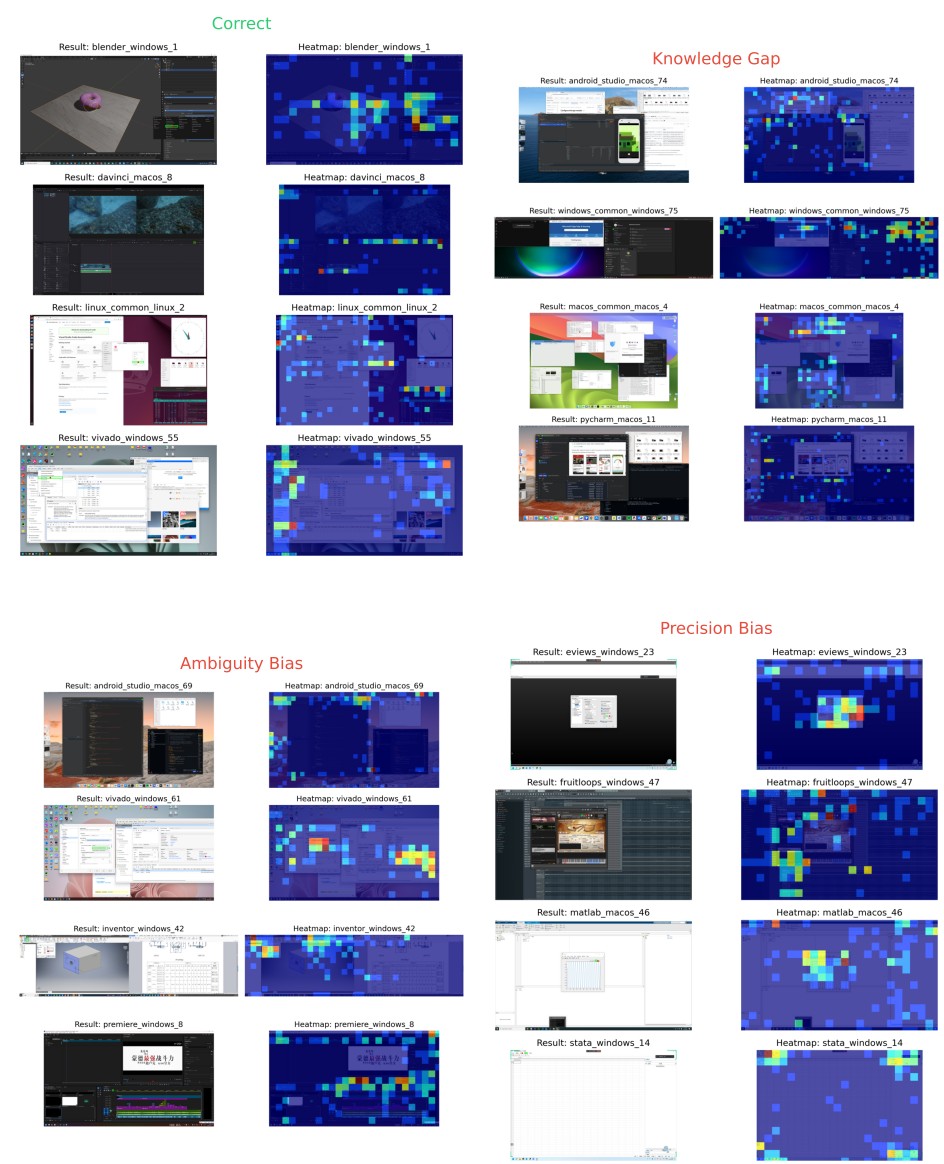

Figure 8: More Attributions Visualizations.

