# OpenReview forum: "ManiCoG: Training-Free Improvement for GUI Grounding via Manipulation Chains"
_ICLR.cc/2026/Conference — ICLR 2026 Conference Withdrawn Submission_

### Official Review · Reviewer_4SEP · 2025-10-16

**Soundness:** 3
**Presentation:** 3
**Contribution:** 3
**Rating:** 4
**Confidence:** 5

**Summary:**

This research addresses the suboptimal performance of existing models on the GUI grounding task in complex scenarios, such as the ScreenSpot-Pro benchmark. It proposes the Masked Prediction Distribution (MPD) attribution method to identify two core sources of error: precision bias caused by high image resolution and ambiguity bias resulting from intricate interface elements.To tackle these issues, the study introduces the ManiCoG (Manipulation-based Chain of GUI Grounding) framework. This framework enhances model performance without requiring additional training by employing two key operations: coarse-to-fine focusing to mitigate precision bias and candidate selection to alleviate ambiguity bias. For example, ManiCoG improves the accuracy of the TianXi-Action-7B model on the ScreenSpot-Pro benchmark from 51.9% to 57.8%.Furthermore, ablation studies have verified the robustness of this method under various parameter configurations.

**Strengths:**

1.Proposed the MPD attribution method to systematically diagnose the three primary sources of error in GUI grounding for the first time: knowledge gap, precision bias, and ambiguity bias.

2.Designed coarse-to-fine focusing and candidate selection operations to specifically mitigate precision bias and ambiguity bias, respectively.

3.Validated the effectiveness of the training-free paradigm by achieving stable improvements across 3 models and 2 benchmarks, offering a new direction for the optimization of the inference stage in GUI grounding.

**Weaknesses:**

1.I noticed that the main body of the paper only presents the experimental results on ScreenSpot-pro, while the results for ScreenSpot-v2 are in the appendix. However, the improvement of ManiCoG on ScreenSpot-v2 is quite limited, with UI-TARS-1.5-7B only achieving a 0.1 increase in average accuracy.

2.The entire process is complex. It first goes through a coarse-to-fine region amplification, followed by a selection among candidate boxes. A prerequisite for this candidate box selection is that the correct answer must be generated during the candidate generation stage, which relies to some extent on randomness.

3.The selection of candidate boxes depends on the reasoning and selection of a more powerful external model. However, a crucial consideration in the field of GUI agents is how to enable the agent to complete tasks faster and better. A two-stage refinement at test-time will significantly increase the agent's decision-making latency, leading to a lot of unnecessary overhead.

**Questions:**

please see weakness. If my concerns are addressed, I will consider raising my score.

---

### Official Review · Reviewer_AR8p · 2025-10-29

**Soundness:** 3
**Presentation:** 3
**Contribution:** 1
**Rating:** 2
**Confidence:** 5

**Summary:**

The authors propose a test-time computation scaling strategy to improve GUI grounding accuracy for GUI agents. Their approach leverages Masked Prediction Distribution (MPD) to analyze failure cases in grounding predictions and introduces two strategies: coarse-to-fine focus and candidate selection. These are evaluated on GUI grounding benchmarks using multiple backbones.

**Strengths:**

- The method is conceptually straightforward and well-motivated.
- The manuscript is well-written and clearly structured.
- The authors provide ablation studies on their two key components, which support their claims empirically.

**Weaknesses:**

- One major concern is that the novelty of the proposed method is highly limited. Unfortunately, the core ideas and findings largely overlap with those of prior work, particularly R-VLM. For instance, R-VLM already adopts a coarse-to-fine grounding strategy and explicitly analyzes precision bias, where the model correctly identifies the target region but exhibits a consistent spatial offset. Furthermore, the ambiguity bias that the authors mentioned is also thoroughly discussed in R-VLM, and closely aligns with the motivation behind its IoU-aware loss function. These conceptual overlaps significantly diminish the originality of the current work. Moreover, regarding the knowledge gap issue, multiple prior works have demonstrated that scaling up pretraining data can effectively mitigate this problem (e.g., OSATLAS, UGround). Similarly, the issue of high-resolution input for accurate GUI understanding has also been emphasized in earlier studies—most notably in CogAgent, which directly addresses this limitation.

- The candidate selection module appears to be the most original component, yet its reliance on hand-crafted heuristics diminishes its impact. If the authors aim to claim originality, they must clearly delineate **how their method differs from existing approaches**. In its current form, Section 2.3 is especially underdeveloped in this regard. A more rigorous positioning within the related literature is strongly recommended.

- Another important limitation is the lack of evaluation on agent tasks. While GUI grounding performance is analyzed, it remains unclear whether improvements in candidate selection or coarse-to-fine focus  translate into downstream agent success (e.g., task completion rates). Evaluation on real-world agent tasks such as AITW, MiniWob, or Mind2Web would strengthen the paper’s practical relevance.

**Questions:**

- One additional point worth considering is whether test-time scaling must be applied uniformly to all test cases. In practice, some instances may already be trivial to resolve without additional computation, and indiscriminate application of scaling strategies may lead to unnecessary overhead. Introducing a selective or adaptive mechanism—for example, based on grounding difficulty—could further enhance the efficiency of the proposed method.

---

### Official Review · Reviewer_Sv89 · 2025-10-31

**Soundness:** 3
**Presentation:** 3
**Contribution:** 2
**Rating:** 4
**Confidence:** 3

**Summary:**

This paper presents ManiCoG, a training-free inference framework designed to improve the accuracy of GUI grounding models. From observed model failure modes, the authors propose a framework to address the main challenges via a coarse-to-fine grounding process. The empirical results are strong, and the method is effective for many GUI grounding models.

**Strengths:**

- Strong motivation. The paper's primary contribution is not just the solution, but the diagnosis of error analysis.

- Good Empirical Results: The method demonstrates clear and consistent performance improvements. The gains are shown to be generalizable across multiple different base models, reinforcing the claim that this is a robust, backbone-agnostic enhancement.

**Weaknesses:**

- Novelty and missing baseline. The method is similar to a method proposed in the ScreenSpotPro benchmark called "ScreenSeeker", which also adopts a coarse-to-fine focusing and selection design. It is confusing that the paper uses the benchmark but ignores the method. What is the main difference between ManiCoG and it?

- What is the cost of ManiCoG, time-wise and cost-wise?

- Potentially unfair comparison: The settings of GUI-RC and DiMo-GUI  are not described in the paper, making it hard to compare between the methods. What planning model is used in the "ManiCoG-7B" result in Table 2? Do GUI-RC and DiMo-GUI also use extra planning models?

**Questions:**

Please refer to the points in Weaknesses.

---

### Note · Authors · 2025-11-12

I have read and agree with the venue's withdrawal policy on behalf of myself and my co-authors.